# Laboratory Exploration of Several Potential Biocontrol Methods Against the Ambrosia Beetle, *Euwallacea interjectus*

**DOI:** 10.3390/insects16010056

**Published:** 2025-01-09

**Authors:** Jialin Liu, Nan Jiang, Haiming Gao, Shengchang Lai, Yang Zhou, Dejun Hao, Lulu Dai

**Affiliations:** 1Co-Innovation Center for Sustainable Forestry in Southern China, College of Forestry, Nanjing Forestry University, Nanjing 210036, China; jialin@njfu.edu.cn (J.L.); jiangnan@njfu.edu.cn (N.J.); lsc@njfu.edu.cn (S.L.); zhouy@njfu.edu.cn (Y.Z.); djhao@njfu.edu.cn (D.H.); 2College of Forestry and Horticulture, Xinjiang Agricultural University, Urumqi 830052, China; gaohaiming0996@163.com

**Keywords:** ambrosia beetle, Coleoptera, Curculionidae, biological control, entomopathogenic microorganisms, ectoparasitic mites

## Abstract

The ambrosia beetle *Euwallacea interjectus*, a pest of economic tree species, has caused significant damage to Chinese poplar plantations since 2020, aided by its symbiosis with *Fusarium populicola*. Current control relies on tree removal and chemicals, with limited biological control options. This study evaluated several potential biocontrol agents, including *Beauveria bassiana*, *Serratia marcescens,* and its metabolite prodigiosin, and the ectoparasitic mites *Pyemotes moseri* and *P. zhonghuajia*. While these agents showed promise in laboratory tests, further investigation is needed for field application.

## 1. Introduction

*Euwallacea interjectus* Blandford (Coleoptera: Curculionidae: Scolytinae) is a fungus-feeding ambrosia beetle that carries mutualistic fungi in specialized mycangia, cultivating them as food within tree xylem [1,2]. These fungi, primarily *Fusarium* species, are pathogenic to trees, causing wilt diseases and, along with the beetles, obstructing vascular tissues, which gradually kills the host tree [2,3,4]. *E. interjectus* is found across 14 Chinese provinces and in the United States, Japan, Argentina, Thailand, Malaysia, and Bangladesh [5,6,7,8,9]. It damages economically important trees, such as poplars in water-stressed regions of Argentina and *Acer negundo* in Florida [6,8]. In Japan, it also vectors *Ceratocystis ficicola*, a fig wilt pathogen [9]. Recent outbreaks in China’s Shanghai and Jiangsu Provinces have devastated over 1500 hectares of poplar stands, killing tens of thousands of trees and threatening local poplar industries [5,10].

The current integrated management of ambrosia beetles includes cultural, chemical, and biological control, as well as the use of semiochemicals [11]. Biological control primarily employs entomopathogenic fungi, such as *Beauveria bassiana* and *Metarhizium anisopliae*, which reduce beetle survival and reproduction in laboratory settings while producing spores to spread infections. Other fungi, such as *Isaria fumosorosea* and *Aspergillus flavus*, also show strong pathogenicity [12,13]. Research on entomopathogenic bacteria is limited, but *Serratia marcescens* has demonstrated insecticidal properties and produces prodigiosin, a compound with potential applications in biological control [14,15]. Antagonistic microorganisms, such as *Trichoderma* spp. and *Bacillus subtilis*, are being studied for their ability to control beetles and disrupt their symbiotic relationships, although further research is needed to confirm these effects [16,17].

Additionally, *Pyemotes* mites (Acariformes: Pyemotidae) are widely distributed ectoparasites, with several species recognized as natural enemies of bark beetles, such as *Pyemotes moseri*, *P. dryas,* and *P. johnmoseri* [18,19,20]. *P. zhonghuajia* and *P. moseri*, both of which were discovered in China [19,21], paralyze and kill their hosts by releasing toxins and then feed on the host’s hemolymph [22]. The opisthosoma of female mites gradually enlarges, producing the next generation of mites. *Pyemotes* mites may also act as vectors for biocontrol fungi, supporting integrated pest management, although their effectiveness against ambrosia beetles requires further investigation [23,24].

To investigate the potential of various biological control agents against the ambrosia beetle *E. interjectus* and its mutualistic fungus, we assessed the efficacy of *B. bassiana* strain B-BB-1, *S. marcescens* strain B-SM-1, the prodigiosin crude extract, and the parasitoids *P. zhonghuajia* and *P. moseri*. Our evaluation focused on their toxicity, antagonistic interactions, and biocontrol potential. This research provides insights into the biocontrol effects of these agents on *E. interjectus* and serves as a reference for their application in managing ambrosia beetles.

## 2. Materials and Methods

### 2.1. Beetle, Strain, and Mite Sources

*Euwallacea interjectus* were collected from poplar trees within Huanghai National Forest Park in Dongtai City, Jiangsu Province, China (120°50′38″ E, 32°52′43″ N). The beetles were reared on an artificial diet at 25 ± 3 °C, and individuals at different life stages were collected. The artificial diet comprised 250–280 g oven-dried Populus sawdust, 10 g each of yeast, casein, and soluble starch, 20 g sucrose, 40 g agar, and 1 L distilled water. The diet preparation and beetle rearing followed Zheng et al. [25].

*Fusarium populicola* was isolated from the mycangia and the inner walls of the galleries of *E. interjectus,* which has been identified as a *Fusarium* mutualist and stored in our laboratory [1]. *Beauveria bassiana* strain B-BB-1 was isolated from naturally diseased and deceased adult female *E. interjectus*. *Serratia marcescens* strain B-SM-1 was isolated from diseased and deceased larvae of *E. interjectus*. Furthermore, the isolates of these two strains were deposited in the China Center for Type Culture Collection (CCTCC) for further study with CCTCC No. M2023270 (B-BB-1) and M2023245 (B-SM-1).

*Pyemotes zhonghuajia* and *P*. *moseri* were sourced from Beijing ShanFeng Biotechnology Co., Ltd. (Beijing, China) They were propagated in a 28 °C incubator under dark conditions using *Tenebrio molitor* pupae as hosts and were subsequently used in the experiments.

### 2.2. Bioassays of B. bassiana and S. marcescens

Since the larvae of *E. interjectus* remain hidden deep within the tunnels before pupation, the spores of *B. bassiana* sprayed on the surface are unlikely to reach the tunnels. Therefore, the application of *B. bassiana* is timed to target the short flight period following the emergence of female adults. The main objective is to investigate whether the infected female adults can die before producing offspring. As a result, the test of *B. bassiana* was conducted only on female adults.

Similarly, regarding *S. marcescens*, unlike the surface-penetrating infection mechanism of fungi, insect pathogenic bacteria typically infect insects through ingestion via the digestive tract. In actual rearing conditions, most infected individuals are larvae and pupae. For the flying female adults, no significant effects of *S*. *marcescens* infection were observed. Therefore, the test for *S. marcescens* was conducted only on larvae.

#### 2.2.1. Bioassay of *B. bassiana* Strain B-BB-1

Petri dish experiments: *B. bassiana* conidia were suspended in sterile water with 0.1% Tween 80 to prepare concentrations of 10^5^, 10^6^, 10^7^, and 10^8^ viable spores/mL. The steps for the production of *B. bassiana* conidia are shown in the Appendix A. Female *E. interjectus* adults were immersed in these suspensions or the control solution (Tween 80 alone) for 30 s, dried, and transferred to Petri dishes lined with moist sterile filter paper (10 beetles per dish, 6 replicates per concentration). The dishes were incubated at 25–28 °C, and survival was monitored daily. Dead beetles were incubated separately to confirm fungal infection. Mortality was corrected using Abbott’s formula [26].Artificial diet experiments: Treated beetles (10^6^ to 10^8^ viable spores/mL or control) were individually placed in 50 mL tubes containing 20 mL of artificial diet (30 replicates/treatment). The reason for treating the beetles with spores instead of the diet is that the high temperature during diet preparation could inactivate the spores. Additionally, the artificial diet is compacted in the tube with a paraffin-sealed surface, making it difficult for spores to penetrate and evenly distribute throughout the diet. After 20 days at 28 °C, the diet was dissected to evaluate parent beetle survival and offspring production. This approach was used as a preliminary model for Populus wood segments, as the artificial diet is easier to dissect for initial assessment.Poplar log experiments: Poplar logs were inoculated with female *E. interjectus* treated with spore powder or spore suspension, along with untreated controls (details in the Appendix A). Each log was inoculated with at least 30 beetles, with a minimum of 10 beetles per treatment, and the experiment was replicated across three logs (Appendix A). Every 2 days, the expelled sawdust was collected, dried, and weighed to assess beetle activity and survival. After 10 days, the logs were dissected to determine beetle survival rates and offspring production.

#### 2.2.2. Bioassay of *S. marcescens* Strain B-SM-1

PDA plates colonized by *F. populicola* were sprayed with five tenfold dilutions of *S. marcescens* broth or sterile water (control). Second- and third-instar *E. interjectus* larvae (10 per plate, 3 replicates per treatment) were introduced to the plates and incubated at 25–28 °C. Larval survival was recorded daily, and dead larvae were removed. Details are provided in the Appendix A.

### 2.3. In Vitro Antagonism of F. populicola by B. bassiana and S. marcescens

The antagonistic effects of *B. bassiana* and *S. marcescens* on *F. populicola* were assessed using dual-culture assays on PDA plates. For *B. bassiana*, the mycelial plugs of each fungus were coinoculated or used individually as controls, and colony growth and inhibition zones were measured. For *S. marcescens*, bacterial streaks were paired with *F. populicola* plugs on PDA plates with similar controls and measurements [16]. The plates were incubated at 28 °C in darkness, and the growth inhibition percentages were calculated [27]. The detailed procedures are provided in the Appendix A.

### 2.4. Toxicity and Antifungal Effects of Prodigiosin Crude Extract (PCE)

To evaluate the toxicity of PCE to *E. interjectus*, PDA plates colonized by *F. populicola* were treated with PCE solution at different concentrations (100%, 10%, 1%, and 0.1%) or sterile water as a control. Ten larvae were introduced per plate, with three replicates per treatment. Larval survival was recorded daily.

For antifungal effects, PDA plates containing 0.1%, 0.2%, and 1% PCE were prepared, with untreated plates serving as controls. *F. populicola* plugs were inoculated at the center, and colony diameters were periodically measured to assess inhibition. Details are provided in the Appendix A.

### 2.5. Preference, Life Cycle, and Parasitism of Pyemotes Species on E. interjectus

For host preference, *E. interjectus* adults, pupae, and larvae were exposed to *Pyemotes* species in sealed 6 cm Petri dishes, and mortality, mite attachment, and opisthosoma condition were recorded daily.

To study the mites’ life cycle, parasitized pupae were placed on filter paper discs in sticky traps, ensuring that all offspring mites were captured. After incubation, offspring were counted using microscopy and ImageJ software (ver. 1.51j8).

For parasitism, *E. interjectus* larvae on *F. populicola*-colonized plates were exposed to *Pyemotes* species, and larval survival, mite attachment, and opisthosoma development were monitored daily. Details are provided in the Appendix A.

### 2.6. Statistical Analysis

For bioassays assessing the effects of *B. bassiana*, *S. marcescens*, and PCE on *E. interjectus* on Petri dishes and mutualistic fungus plates, survival curves were analyzed using the Kaplan–Meier method with log-rank tests and Bonferroni-corrected pairwise comparisons (*p* < 0.05). In assays involving *B. bassiana* on artificial diets and in poplar logs, survival rates and offspring production under different treatments were compared using chi-square tests and Bonferroni adjustments.

Data not meeting normality or homogeneity of variance assumptions, including offspring production in artificial diets, the dry weight of expelled sawdust from logs, and the relative quantity of *Pyemotes* species on *E. interjectus* developmental stages, were analyzed using the Kruskal–Wallis test, followed by Dunn–Bonferroni pairwise comparisons. For the antagonistic effects of *B. bassiana* and *S. marcescens* against *F. populicola*, the antifungal efficacy of PCE, and differences in the results between the two *Pyemotes* species, independent-sample t-tests were applied, as these data were normally distributed.

All the data, except for the log-rank test, which was completed with Origin 2022 (OriginLab Corporation, Northampton, MA, USA), were analyzed using SPSS 26 (IBM Corporation, Armonk, NY, USA). The figures were plotted with Origin 2022 (OriginLab Corporation, Northampton, MA, USA).

## 3. Results

### 3.1. Lethal Effect of B. bassiana and S. marcescens to E. interjectus

#### 3.1.1. Lethal Effect of *B. bassiana* to *E. interjectus*

The results of the Petri dish experiment indicated that the survival curves of *E. interjectus* differed markedly following exposure to different concentrations of *B. bassiana* spore suspensions (Figure 1a). Statistical significance among the groups was confirmed by the log-rank test (*χ*^2^ = 131.45, df = 4, *p* < 0.001). The pairwise comparison results revealed that, except for the 10^5^ spores/mL treatment, which did not differ significantly from the control (*χ*^2^ = 0.28, df = 1, *p* > 0.05), all the other treatments presented significant differences from the control. Overall, mortality increased with increasing spore concentration. However, no significant difference was observed between the 10^7^ and 10^8^ spores/mL treatments (*χ*^2^ = 6.40, df = 1, *p* > 0.05), indicating that concentrations above 10^7^ spores/mL were sufficient to achieve effective results.

After 20 days, dissection of the artificial diets revealed that female survival in the 10^6^ spores/mL treatment group was 36.67 ± 8.80% (n = 30), which was significantly lower than that in the control group (83.33 ± 6.80%; n = 30). The 10^7^ and 10^8^ spores/mL treatments resulted in complete mortality (Table 1). These results corroborate the findings from the Petri dish experiments. Although the 10^7^ spores/mL treatment was lethal, 23.33 ± 7.72% (n = 30) of the females still produced offspring, with some exceeding 30. Conversely, the 10^8^ spores/mL treatment limited reproduction, with only 3.33 ± 3.28% (n = 30) producing fewer than 10 offspring, indicating superior efficacy (Table 1, Figure 1b).

Dry weight measurements of the sawdust expelled by the beetles revealed that most beetles in the control group remained active, with only a brief plateau period between 4 and 6 days, and continued to expel sawdust throughout the experiment. In contrast, the beetles in the two treatment groups experienced negligible sawdust expulsion after the fourth day, suggesting that they were either nearly dead or had died by this time (Figure 1c). This finding matched the timing of mass mortality observed in the Petri dish experiments. To confirm the survival status of the beetles, the logs infested for 10 days were split open, revealing a survival rate of 71.08 ± 7.77% (n = 39) for the control group females, whereas no females survived in the two treatment groups, indicating a significant lethal effect of the *B. bassiana* treatment (chi-square test, *p* < 0.001). Among the treated beetles, 44.18 ± 11.78% (n = 42) and 75.08 ± 6.45% (n = 44) emerged and died at the bottoms of the collection tubes (spore suspension and powder, respectively), with others found blocking gallery entrances (Appendix A).

#### 3.1.2. Lethal Effect of *S. marcescens* to *E. interjectus*

Using the plate spreading method, the highest treatment concentration was determined to be approximately 1.6 × 10^9^ cfu/mL. The survival curves of *E. interjectus* larvae treated with varying concentrations substantially overlapped, with no distinct mortality peaks (Figure 2a). Log-rank tests revealed statistically significant differences between groups (*χ*^2^ = 25.45, df = 5, *p* < 0.001). The pairwise comparison results indicated that there were no significant differences among all the treatments (*p* > 0.05), and even the second-highest concentration treatment (1.6 × 10^8^ cfu/mL) resulted in no significant difference compared with the control (*χ*^2^ = 8.00, df = 1, *p* > 0.05). *S. marcescens* showed a generally moderate and inconsistent lethal effect but caused larval mortality (Figure 2b), with some larvae ingesting sufficient bacteria to result in death only after pupation (Figure 2c).

### 3.2. The Antagonistic Effects of B. bassiana and S. marcescens on F. populicola

#### 3.2.1. *B. bassiana* Versus *F. populicola*

A continuous recording of the radius of each colony revealed that from day 7 onward, the growth of *F. populicola* on competition plates was significantly inhibited (*t* = 5.515, df = 10, *p* < 0.001), with a growth inhibition percentage of 11.06%. This inhibition occurred before the mycelia of the two fungi were in contact. In contrast, *B. bassiana* showed no significant inhibition until day 10, at which point a more pronounced inhibitory effect was observed (*t* = 2.391, df = 10, *p* = 0.038), with a growth inhibition percentage of 17.90%. By this time, the percentage of growth inhibition for *F. populicola* had reached 34.95%. The width of the inhibition zone between the two mycelia was 8.00 ± 0.34 mm (n = 6), which was often accompanied by dark pigmentation along the edge of the *F. populicola* colony (Figure 3b).

#### 3.2.2. *S. marcescens* Versus *F. populicola*

Commencing on the third day, the colony radius of *F. populicola* on the competition plates was significantly different from that of the control group (*t* = 11.578, df = 9, *p* < 0.001), with a growth inhibition percentage of 22.10%. Mycelial growth continued to be significantly inhibited but never ceased, maintaining a stable inhibition percentage and making contact with *S. marcescens* colonies on the fifth day (Figure 4b). No formation of inhibition zones between the two species was noted, nor was any pigmentation detected. *F. populicola* was able to grow through the *S. marcescens* colony, whereas the latter permeated the interstices of the former’s mycelium.

### 3.3. Toxicity and Antifungal Effects of PCE

Larvae reared on *F. populicola* plates treated with varying concentrations of PCE presented distinct survival curves (Figure 5a). The results of the Kaplan–Meier log-rank test indicated significant differences between groups (*χ*^2^ = 190.58, df = 4, *p* < 0.001). The larvae in the 100% PCE treatment group reached a corrected mortality rate of 86.11 ± 8.18% (n = 36) within 24 h, with complete mortality observed by 48 h, demonstrating the highly toxic effects of high concentrations of PCE on the larvae. In contrast, the 10% PCE treatment group had a corrected mortality rate of only 2.53 ± 6.16% (n = 36) after 24 h, increasing to 36.11 ± 14.87% after 48 h.

The measurement of colony diameter revealed that *F. populicola* in the three treatment groups was continuously suppressed throughout the experimental period. The presence of PCE had a pronounced inhibitory effect on the growth of *F. populicola*, with increased efficacy observed at higher concentrations. Even the treatment group with a mere 0.1% PCE concentration demonstrated a significant inhibition percentage of 21.40% on day 5, with the colony diameter substantially reduced compared with that of the control (*t* = 24.872, df = 17, *p* < 0.001). In contrast, the 1% PCE treatment group achieved an inhibition percentage of 52.23% at this time, which was also significantly lower than that of the control (*t* = 4.671, df = 12.236, *p* < 0.001) (Figure 5b).

### 3.4. Preferences of the Two Pyemotes Species at Different Stages of E. interjectus Development

Due to the uncontrollable release of mites, there were apparent differences in total mite counts across groups, so relative distributions were utilized in lieu of absolute numbers for comparative purposes. The patterns of the relative distributions of *P. moseri* and *P. zhonghuajia* across different developmental stages of *E. interjectus* exhibited broadly similar trends (Figure 6). After 24 h, the predominant distribution of mites was observed on the larvae and pupae, with a slightly lower number on the larvae than on the pupae. Starting on the second day, some larval carcasses began to show signs of dehydration and shrinkage or infection, and the mites on their surfaces started to disperse. Overall, compared with larvae and adults, beetle pupae have advantages such as reduced water loss, being cleaner and less prone to microbial contamination, lack of a hard exoskeleton, a larger surface area, and higher nutrient content, making them easier to parasitize. Consequently, the two *Pyemotes* species showed a preference for pupae, where their opisthosomata developed best and in the greatest numbers (Table 2).

### 3.5. Life Cycles of the Two Pyemotes Species in the Parasitization of E. interjectus

Continuous observation and photographic documentation revealed that the life cycles of the two *Pyemotes* species were largely consistent (Figure 7). After 1 day, the pupae were densely covered with mites, at which time the majority of the mites’ opisthosomata had not yet initiated development. By the second day, the opisthosomata of most mites had begun to develop and expand. On the third day, the opisthosomata had attained their final size. Between days 6 and 7, mature male offspring began emerging and remained around the mother’s genital opening. After the seventh day, the female offspring began to appear, mating with the males before disseminating to seek new hosts. The mother’s opisthosoma started to cave in and rupture, with the dissemination process capable of lasting for more than a week. By counting the disseminated offspring mites, no significant differences were noted between the two *Pyemotes* species with respect to the number of opisthosomata formed on individual female pupae of *E. interjectus*, the number of offspring produced, or the average number of offspring per opisthosoma (Table 3).

### 3.6. Parasitism of Two Pyemotes Species on E. interjectus

Both *P. moseri* and *P. zhonghuajia* successfully located and parasitized *E. interjectus* larvae on mutualistic fungus plates, with 90 ± 5.00% (n = 40) and 90 ± 3.53% (n = 40) of the larvae infested within one day, resulting in 100% mortality, even if there was only one mite on their surface. However, by day three, larval carcasses were fully colonized by bacteria or fungi, leading to mite death and incomplete opisthosomata development, preventing either species from completing their life cycle (Appendix A).

## 4. Discussion

As one of the most widely used entomopathogenic fungi, *Beauveria bassiana* has been reported to effectively kill numerous species of beetles in Scolytinae, such as *Dendroctonus ponderosae*, *Dendroctonus rufipennis*, *Xylosandrus germanus*, *Xyleborus affinis*, and *Xylosandrus crassiusculus*. It has been shown to effectively kill these beetles and potentially inhibit the growth of mutualistic fungi within beetle tunnels, thereby slowing the population growth of these ambrosia beetles [28,29,30]. In this study, *B. bassiana* demonstrated a significant lethal effect on female adults of *Euwallacea interjectus*. These findings suggest the potential of *B. bassiana* as a biocontrol agent against *E. interjectus*, which aligns with previous reports on other beetle species [31,32]. Additionally, confrontation assays on plates revealed that *B. bassiana* could inhibit the growth of *F. populicola*, which is consistent with the findings of Prabhukarthikeyan et al. [33], potentially disrupting the mutualistic relationship between beetles and their fungi, thereby affecting offspring development [16].

*Serratia marcescens* is a common bacterium with known pathogenicity against insects such as *Phyllophaga blanchardi* and *Curculio dieckmanni* [15,34]. However, reports linking *S. marcescens* to Scolytinae beetles are limited and involve species such as *Hylurgus ligniperda*, *Dendroctonus frontalis*, and *Platypus quercivorus* [35,36,37]. In this study, *S. marcescens* had weak and inconsistent lethal effects on *E. interjectus* larvae, likely due to its low virulence. Several factors may also have contributed to these variations. First, uneven colonization of the mutualistic fungus on the plates or larval feeding behavior could lead to areas with thinner hyphae. Due to the rapid growth of *S. marcescens*, droplets of bacterial colonies may form in these areas, which larvae can pick up, disrupting the intended concentration gradient and causing significant variation between replicates. Additionally, the relatively short larval stage may have led to some larvae pupating during the experiment, which could affect feeding behavior and result in variability in bacterial intake. Moreover, observed cannibalism among larvae, where weaker or dead larvae are consumed, may facilitate the spread of *S. marcescens* between individuals.

Despite some studies indicating that *S. marcescens* can inhibit *Fusarium* species by producing chitinase, which degrades fungal cell walls [38], our plate confrontation assays revealed no inhibition zones between *S. marcescens* and *F. populicola*, with *F. populicola* even growing through bacterial colonies. This could be due to the use of a PDA medium, which favors fungal growth. Alternatively, the inhibitory capacity of *S. marcescens* against *F. populicola* might be inherently weak, with the reduced diameter of *F. populicola* potentially attributable to nutrient competition on the plate.

As a key metabolite of *S. marcescens*, prodigiosin has been reported to have insecticidal and antifungal activities, such as reducing the survival and pupation rates of *Aedes aegypti* and inhibiting oviposition and egg hatching in *Diaphorina citri* [39,40]. It also inhibits several plant pathogenic fungi, such as *Colletotrichum* spp., *Pythium myriotylum*, and *Rhizoctonia solani* [41,42,43]. Despite the limited purity of the prodigiosin extracted in our experiments due to constraints in preparation techniques, the crude extract showed significant toxicity to *E. interjectus* larvae and strong inhibition of *F. populicola*, highlighting its potential in biological control.

Although these entomopathogenic microorganisms have shown promising results under laboratory conditions, replicating such success in the field remains challenging due to various environmental factors and the beetles’ specific lifestyle. Effective application requires careful selection of weather conditions and the use of protective agents, such as humic acid, sesame oil, or colza oil, to shield *B. bassiana* from UV damage and prolong its efficacy [44,45]. Furthermore, accurate monitoring of beetle populations is essential for optimizing application timing. Additionally, while *S. marcescens* is moderately effective, its opportunistic pathogenicity poses potential risks to human health, making it a less ideal control option. Prodigiosin, although promising as a biopesticide, faces limitations because of its high production cost and the logistical challenges associated with delivering treatments into beetle galleries. Further research and optimization are needed to achieve field applicability.

Several *Pyemotes* mites have been recognized as natural enemies of bark beetles. For example, *P. moseri*, *P. dryas*, and *P. johnmoseri* have been reported to attack *Cryphalus* spp., *Pityophthorus annectans*, and *Hypoborus ficus*, respectively [18,19,20]. In this study, *P. moseri* and *P. zhonghuajia* were shown to be able to locate and kill *E. interjectus*. Moreover, *Pyemotes* mites can locate pests in tunnels effectively. For example, *P. moseri* controls *Anoplophora glabripennis* by entering larval tunnels [23], and *P. zhonghuajia* targets *Phthorimaea operculella* within potatoes [46]. However, *E. interjectus* differs from these cases. Although the mites can locate and parasitize larvae and pupae in underexposed conditions, they struggle to penetrate the narrow and sealed galleries of *E. interjectus* and exhibit poor parasitism of adult female beetles. Confined gallery spaces, mutualistic fungi, and challenges in dispersal may limit their efficacy. Furthermore, the specificity of *Pyemotes* mites for *E. interjectus* remains uncertain, raising concerns about host switching and potential risks to humans, as some species can cause dermatitis. These factors warrant careful evaluation before large-scale mite release.

The biological control of ambrosia beetles remains a global challenge. Although this study suggests several potential control measures for *E. interjectus*, the unique survival strategy of the beetle, which involves sealing gallery entrances with expelled sawdust, complicates this issue. This behavior not only blocks contamination by extraneous pathogens but also prevents the entry of natural enemies, increasing the difficulty of control efforts and raising the bar for existing biological control methods. To meet this challenge, researchers and forestry managers must continually innovate and seek new strategies and methods to overcome this natural barrier.

## 5. Conclusions

In this study, we assessed the effects of *Beauveria bassiana* strain B-BB-1, *Serratia marcescens* strain S-SM-1, crude extract of prodigiosin from *S. marcescens*, *Pyemotes zhonghuajia*, and *Pyemotes moseri* on *Euwallacea interjectus* and its mutualistic fungus *Fusarium populicola*. *B. bassiana* significantly increased mortality in female *E. interjectus* adults and inhibited *F. populicola* growth. *S. marcescens* was lethal to *E. interjectus* larvae but with inconsistent efficacy, and it showed weak inhibition of *F. populicola*. Prodigiosin extract exhibited high toxicity to *E. interjectus* larvae and significantly inhibited *F. populicola*. Both *P. zhonghuajia* and *P. moseri* attacked *E. interjectus*, preferentially targeting pupae, and exhibited locating and killing capabilities against the beetle larvae on mutualistic fungus plates. These agents show promise for biocontrol, but challenges remain in delivering them into beetle galleries within trees, necessitating further research.

## Figures and Tables

**Figure 1 insects-16-00056-f001:**
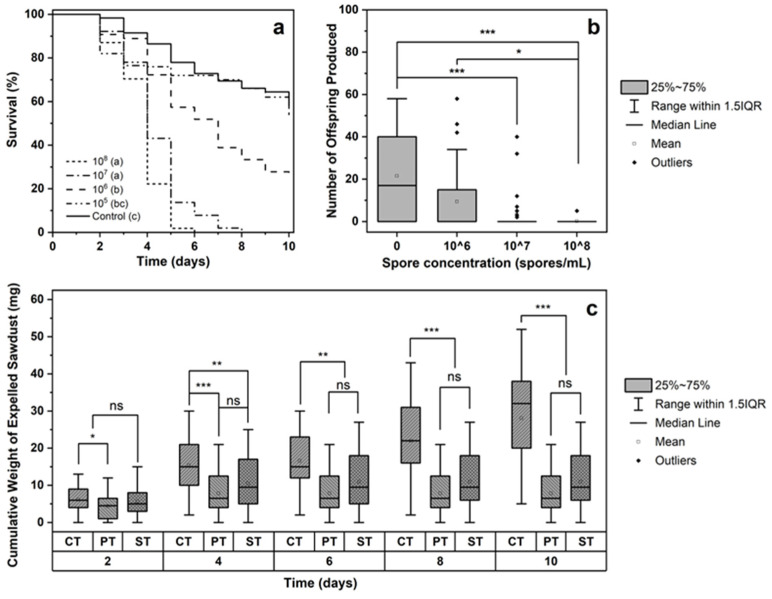
Effects of *B. bassiana* treatments on *E. interjectus* survival, reproduction, and sawdust expulsion. (**a**) Survival curves for female adults of *E. interjectus* treated with different concentrations of spore suspensions of *B. bassiana* in Petri dishes. Curves followed by different letters were significantly different (log-rank test, *p* < 0.05). (**b**) Offspring production of female *E. interjectus* adults treated with different concentrations of *B. bassiana* spore suspensions and control in artificial diet after 20 days (Dunn–Bonferroni test; significant differences are shown by asterisks: * *p* < 0.05, *** *p* < 0.001). (**c**) Cumulative weight of expelled sawdust in poplar log experiment under three treatment methods. “CT” represents “control treatment,” “PT” represents “spore powder treatment”, and “ST” represents “spore suspension treatment” (Dunn–Bonferroni test; significant differences are shown by asterisks: ns = not significant results, * *p* < 0.05, ** *p* < 0.01, *** *p* < 0.001).

**Figure 2 insects-16-00056-f002:**
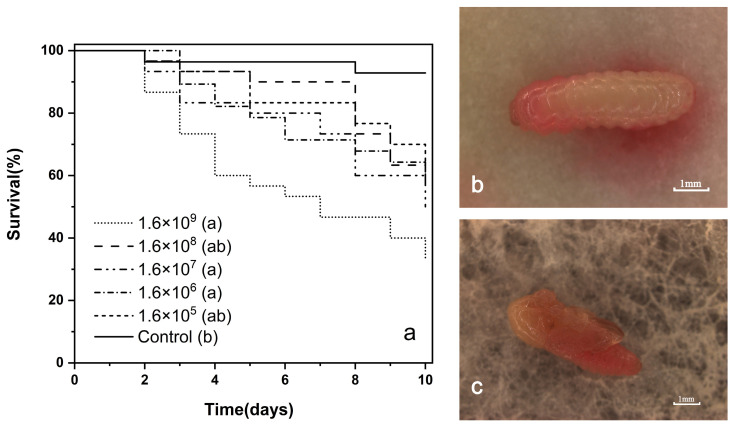
Effects of *S. marcescens* on *E. interjectus* survival. (**a**) Survival curves for larvae of *E. interjectus* on the mutualistic fungi plates treated with different concentrations of *S. marcescens*. Curves followed by different letters were significantly different (log-rank test, *p* < 0.05). (**b**) Dead larva of *E. interjectus* infected by *S. marcescens*. (**c**) Dead pupa of *E. interjectus* infected by *S. marcescens*.

**Figure 3 insects-16-00056-f003:**
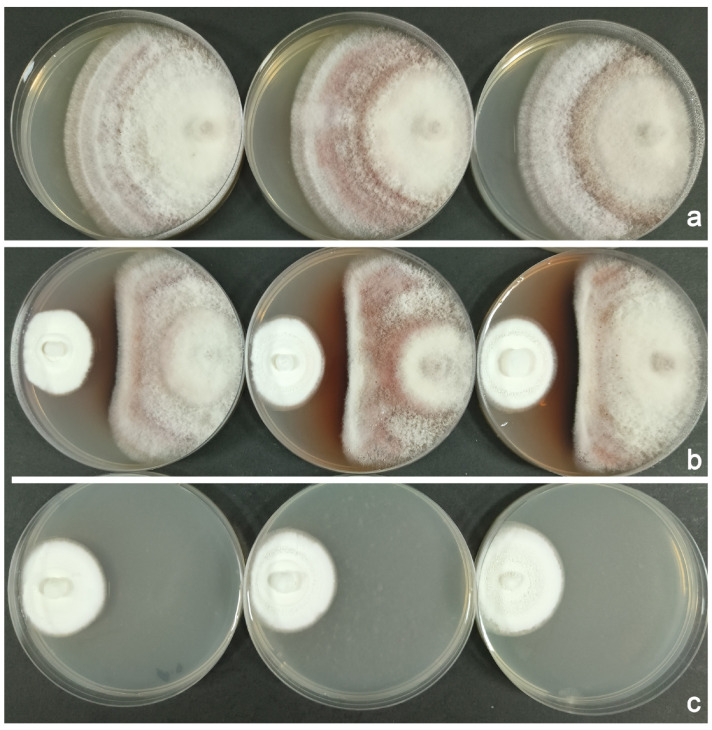
Examples of in vitro interactions between *F. populicola* and *B. bassiana*. (**a**) *F. populicola* control plates. (**b**) Inhibition zones formed between *F. populicola* (inoculated on the right of the plate) and *B. bassiana* (inoculated on the left of the plate). (**c**) *B. bassiana* control plates.

**Figure 4 insects-16-00056-f004:**
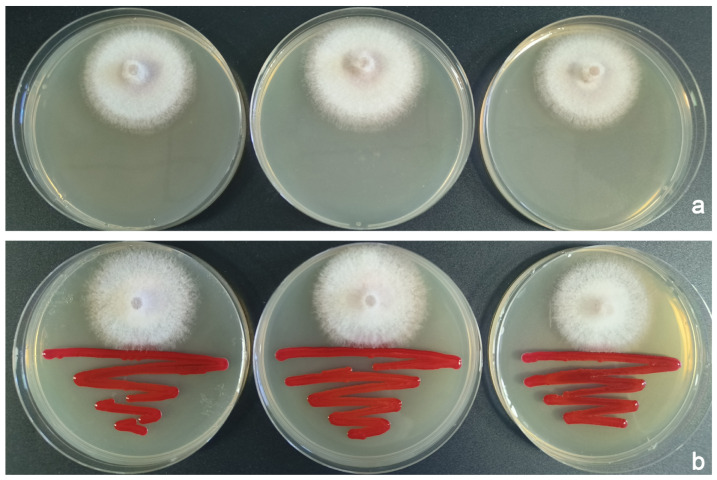
Examples of in vitro interactions between *F. populicola* and *S. marcescens*. (**a**) *F. populicola* control plates. (**b**) Competition plates between *F. populicola* (inoculated on the top of the plate) and *S. marcescens* (inoculated on the bottom of the plate).

**Figure 5 insects-16-00056-f005:**
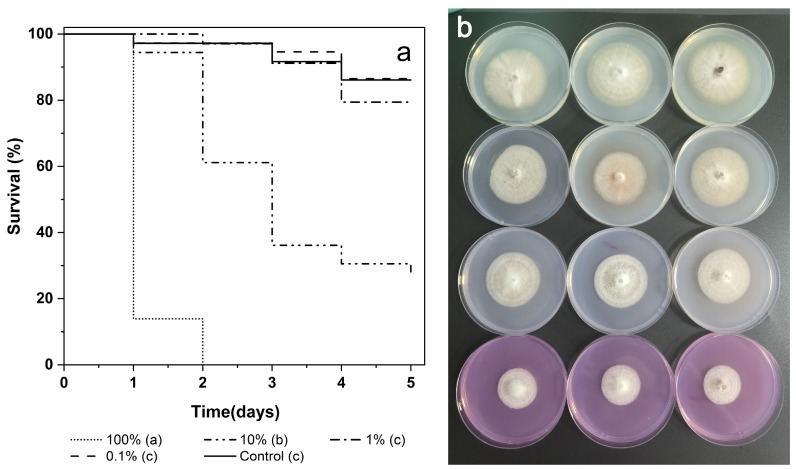
Effects of PCE concentrations on *E. interjectus* survival and *F. populicola* growth. (**a**) Survival curves for larvae of *E. interjectus* on the mutualistic fungi plates treated with different concentrations of the PCE. Curves followed by different letters were significantly different (log-rank test, *p* < 0.05). (**b**) Colony growth of *F. populicola* cultures on PDA plates containing different concentrations of PCE after 5 days. From top to bottom: control, 0.1% PCE treatment, 0.2% PCE treatment, 1% PCE treatment.

**Figure 6 insects-16-00056-f006:**
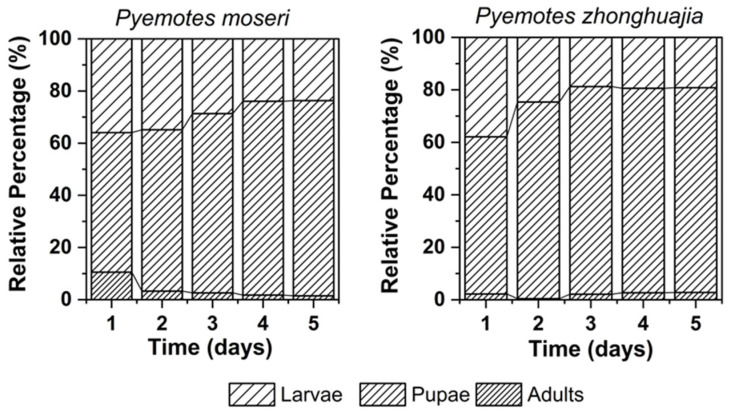
Relative distribution changes in two *Pyemotes* species on different stages of *E. interjectus*. On the first day, the total number of mites on the surface of the beetles was counted. From the second day onward, the number of mites with developing opisthosoma on the beetle surface was recorded.

**Figure 7 insects-16-00056-f007:**
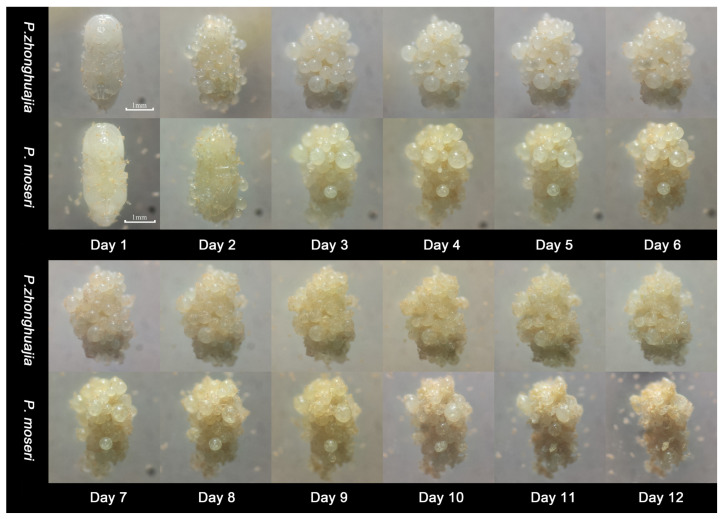
Life cycle of two *Pyemotes* species on the female pupae of the *E. interjectus*.

**Table 1 insects-16-00056-t001:** Survival and offspring production of female *E. interjectus* adults treated with *B. bassiana*.

Treatment	Survival (Mean ± SE%)	Offspring Production (Mean ± SE%)
10^8^	0.00 ± 0.00 ^c^	3.33 ± 3.28 ^d^
10^7^	0.00 ± 0.00 ^c^	23.33 ± 7.72 ^bc^
10^6^	36.67 ± 8.80 ^b^	43.33 ± 9.05 ^b^
Control	83.33 ± 6.80 ^a^	70.00 ± 8.37 ^a^

The data were collected on day 20 from the artificial diet. Values not followed by the same letter in a column are significantly different (chi-square test with Bonferroni correction, *p* < 0.05).

**Table 2 insects-16-00056-t002:** Relative distribution of opisthosoma on *E. interjectus* infested by two *Pyemotes* species.

Treatment	The Relative Quantity of the Opisthosoma			
	(Mean ± SE%)			
	Adults	Pupae	Larvae	
*P. moseri*	1.45 ± 0.71 ^b^	74.84 ± 7.46 ^a^	23.71 ± 7.85 ^ab^	*H* = 10.898, df = 2, *p* = 0.004
*P. zhonghuajia*	2.77 ± 1.17 ^b^	78.05 ± 5.35 ^a^	19.17 ± 5.58 ^ab^	*H* = 10.674, df = 2, *p* = 0.005
	*t* = −0.866, df = 8, *p* = 0.412	*t* = −0.313, df = 8, *p* = 0.762	*t* = 0.421, df = 8, *p* = 0.685	

The data were collected on day 5. Independent-sample t-tests were used to compare the two *Pyemotes* species within each column. Kruskal–Wallis tests followed by Dunn–Bonferroni pairwise comparisons were used to compare the three developmental stages within each row, with different letters indicating significant differences in the results of pairwise comparisons (*p* > 0.05).

**Table 3 insects-16-00056-t003:** Metrics of single female pupae of *E. interjectus* parasitized by two *Pyemotes* species.

Treatment	Number of Opisthosomata (Mean ± SE)	Total Number of Offspring (Mean ± SE)	Number of Offspring per Opisthosoma (Mean ± SE)
*P. moseri*	50.10 ± 2.23	549.10 ± 76.89	10.96 ± 1.58
*P. zhonghuajia*	52.70 ± 3.57	707.00 ± 97.60	13.42 ± 2.68
	*t* = −0.585, df = 18, *p* = 0.566	*t* = −1.206, df = 18, *p* = 0.244	*t* = −1.094, df = 18, *p* = 0.288

Independent-sample *t*-tests were used to compare the two *Pyemotes* species within each column.

## Data Availability

The data presented in this study are available upon request from the corresponding author.

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
