# Peer review of "Laboratory Exploration of Several Potential Biocontrol Methods Against the Ambrosia Beetle, Euwallacea interjectus"

_insects, 2025, doi:10.3390/insects16010056_

Round 1

Reviewer 1 Report

Comments and Suggestions for Authors

The study by Liu et al. evaluated the efficacy of Beauveria bassiana, Serratia marcescens, its metabolite prodigiosin, and two Pyemotes mite species as a potential biocontrol agent for Euwallacea interjectus. The experiment showed promising results in controlling beetle populations. Despite promising results, the authors also highlighted the challenges in the field application of these control agents, including delivery into beetle galleries and potential ecological risks.

Overall, the manuscript is well-structured, and the experiments are methodically conducted. The data analysis is thorough, and the conclusions drawn are well-supported by the results. The figures are presented clearly, making the findings easily accessible to readers.

Here are a few minor suggestions to further improve the manuscript:

Line 55: Spell out the genus name at the beginning of a sentence.

Line 81: Genus name on Pyemotes moseri can be abbreviated.

Methodology:  The authors used various life stages of beetles to test the efficacy of selected biocontrol agents. However, I find a gap in providing the rationale behind selecting specific developmental stages with specific BC agents. For example, why were adult beetles but not larval stages used for B. bassiana?

Line 95-98: I wonder why the authors chose to treat beetles with spores and feed on an artificial diet instead of feeding beetles with an artificial diet treated with different concentrations of spores.

Author Response

Comment 1:Line 55: Spell out the genus name at the beginning of a sentence.

Response 1: We have revised it in new version manuscript.

Comment 2: Line 81: Genus name on Pyemotes moseri can be abbreviated.

Response 2: We have revised it in new version manuscript.

Comment 3: Methodology: The authors used various life stages of beetles to test the efficacy of selected biocontrol agents. However, I find a gap in providing the rationale behind selecting specific developmental stages with specific BC agents. For example, why were adult beetles but not larval stages used for B. bassiana?

Response 3: This design is based on the life habits of the beetle and the infection mechanisms of the selected biocontrol agents. Since the larvae of Euwallacea interjectus remain hidden deep within the tunnels before pupation, the spores of Beauveria bassiana sprayed on the surface are unlikely to reach the tunnels. Therefore, the application of B. bassiana is timed to target the short flight period following the emergence of female adults. The main objective is to investigate whether the infected female adults can die before producing offspring. As a result, the toxicity test of B. bassiana was conducted only on female adults.

Similarly, regarding Serratia marcescens, unlike the surface-penetrating infection mechanism of fungi, insect pathogenic bacteria typically infect insects through ingestion via the digestive tract. In actual rearing conditions, most infected individuals are larvae and pupae. For the flying female adults, feeding is almost absent, and previous tests of high-concentration S. marcescens immersion on adult females showed negligible infection effects. Therefore, the toxicity test for S. marcescens was conducted only on larvae.

Comment 4: Line 95-98: I wonder why the authors chose to treat beetles with spores and feed on an artificial diet instead of feeding beetles with an artificial diet treated with different concentrations of spores.

Response 4: Firstly, this decision is related to the state and preparation process of the diet. Before the artificial diet is prepared, it is in a high-temperature state. Unlike chemical pesticides, adding spores at this stage could result in inactivation due to the heat, thus affecting the experimental outcomes. Additionally, the artificial diet is similar to a moist cork, tightly compacted at the bottom of the plastic centrifuge tube, with the surface sealed with a paraffin layer. Simply adding spore suspension to the surface would not allow it to penetrate or evenly distribute throughout the diet.

Furthermore, this approach is related to the main objective of our study. Due to the hardness of the Populus wood segments, which makes dissection difficult, the artificial diet serves as a simplified model for the wood segments. It is used to preliminarily determine whether infected female adults can die before producing offspring after infection.

Reviewer 2 Report

Comments and Suggestions for Authors

The ms “Laboratory exploration of several potential biocontrol methods against the ambrosia beetle, Euwallacea interjectus” evaluated several natural enemies against a forest pest bark beetle and its symbiont. Though all the experiments were conducted in lab, it could still give some clues in the field of control strategy of the target beetle. The whole ms. was well organized and written so it can be published after make some revision.  Please see the below:

Line 72: artificial diet should be given more detailed information.

Line 285: Do you really mean “lower susceptibility to infection ”? if that is true, then the pupae should be lower parasitized.

Line 297-298: I think the authors want to describe “with different letters indicating significant different in results of pairwise comparisons”

Line 342-344: since S. marcescens is a bacterium, so I do not know why the author refer fungal here, also how larval behavior was related here? Furthermore, “cannibalism among larvae”?  If the author did not know much about these relationships and lack the knowledge about the behavior of beetle, he/she should not discuss it here.

Author Response

Comment 1: Line 72: artificial diet should be given more detailed information.

Response 1: The artificial diet consisted of 250–280 g of oven-dried Populus sawdust (12-mesh), 10 g each of active dry yeast, casein, and soluble starch, 20 g of sucrose, 40 g of agar, and 1 L of distilled water. All components except sawdust were boiled in an aluminum pot with constant stirring until the agar dissolved. Sawdust was then added in portions and thoroughly mixed. The mixture was packed into 50 mL plastic centrifuge tubes, compacted with a wooden rod, wrapped with kraft paper, and sterilized at 121℃ for 20 minutes. In a sterile environment, the diet was further compacted to ~20 mL per tube using a sterile wooden rod, cooled, and air-dried for 18 hours to reduce moisture. To prevent excessive evaporation or surface contamination, 0.7 mL of paraffin was added to seal the diet surface, and the tubes were capped tightly for storage.

For subculturing, the paraffin was pierced, and a single mated female adult was introduced into each tube, which was then capped to prevent escape. After 2 days, successful tubes (with evidence of boring) had their caps replaced with 100-mesh polyethylene screens to allow ventilation and prevent surface microbial growth. After ~30 days of incubation at 28℃, a large number of mated female offspring emerged, enabling further subculturing or experimental use.

Comment 2:Line 285: Do you really mean “lower susceptibility to infection ”? if that is true, then the pupae should be lower parasitized.

Response 2: What I intended to express is that, during the "prepupal" stage before pupation in holometabolous insects, feeding stops, and the insect clears its digestive tract of contaminants and waste in preparation for pupation. Therefore, compared to larvae, pupae are cleaner, and when they are parasitized by Pyemotes mites and die, they are less likely to be contaminated by microorganisms, unlike larvae. Of course, this explanation is merely a hypothesis I have regarding the observed phenomenon.

Comment 3: Line 297-298: I think the authors want to describe “with different letters indicating significant different in results of pairwise comparisons”

Response 3: We have changed the sentence as your suggestion, thank you.

Comment 4: Line 342-344: since S. marcescens is a bacterium, so I do not know why the author refer fungal here, also how larval behavior was related here? Furthermore, “cannibalism among larvae”?  If the author did not know much about these relationships and lack the knowledge about the behavior of beetle, he/she should not discuss it here.

Response 4: The "fungus" referred to here is the mutualistic fungus of the beetle. For the mutualistic fungal plates that were inoculated, the fungal hyphae do not always cover the plates 100%, and the larval behavior (referring to the larvae's feeding and movement) can cause areas of the plate to have thinner or completely exposed hyphae (mainly around the edges of the medium). In these areas, droplets of Serratia marcescens colonies may grow. When larvae approach these areas, they can pick up large amounts of S. marcescens, thereby disrupting the intended concentration gradient of the experiment. The "cannibalism among larvae" is an observed phenomenon in both rearing and experiments. The larvae of this beetle do exhibit behavior where they feed on weaker or dead conspecifics. If one larva feeds on another that has been infected and killed by S. marcescens, it could lead to significant spread of S. marcescens between individuals.